# Inhibition Effect of Hydrophobic Functional Organic Corrosion Inhibitor in Reinforced Concrete

**DOI:** 10.3390/ma15207124

**Published:** 2022-10-13

**Authors:** Jinzhen Huang, Jie Hu, Jinshun Cai, Haoliang Huang, Jiangxiong Wei, Qijun Yu

**Affiliations:** 1School of Materials Science and Engineering, South China University of Technology, Guangzhou 510640, China; 2State Key Laboratory of High Performance Civil Engineering Materials, Jiangsu Research Institute of Building Science, Nanjing 211108, China; 3College of Civil Engineering, Hefei University of Technology, Hefei 230009, China

**Keywords:** corrosion inhibitor, electrochemical performance, pore structure, contact angle, capillary water absorption, reinforced concrete

## Abstract

Using an admixed organic corrosion inhibitor is one of the most efficient strategies to enhance the corrosion resistance and durability of reinforced concrete. However, traditional admixed organic corrosion inhibitors only increase the corrosion resistance of the embedded reinforcing steel, and the optimization effect on the pore structure and the impermeability of concrete is very limited. In this study, in order to evaluate the corrosion-inhibition effect of a novel hydrophobic functional organic corrosion inhibitor, the adsorption behavior of a hydrophobic functional organic corrosion inhibitor and its related effect on the electrochemical behavior of the reinforcing steel was investigated. In addition, this paper further discusses the effect of a hydrophobic functional organic corrosion inhibitor on pore structure and hydrophobic properties, as well as the impermeability of concrete. The results indicated that the hydrophobic functional organic corrosion inhibitor was effectively adsorbed on the surface of the steel bar, and the higher adsorption content was relevant to the higher inhibitor dosage. On one hand, the hydrophobic functional organic corrosion inhibitor exhibited both a pore-blocking effect and a hydrophobic effect on concrete, leading to a refined pore structure and reduced capillary water absorption amount; on the other hand, the hydrophobic functional organic corrosion inhibitor exhibited an excellent corrosion-inhibition effect on the reinforcement embedded in the concrete, presenting an inhibition efficiency higher than 90% with a concentration of 4 wt.%.

## 1. Introduction

Reinforced concrete is the most widely used civil engineering material in marine environments. However, due to its porous microstructure, aggressive chlorides exhibiting small ionic radii and high chemical activity in seawater can transport through the concrete and reach the surface of reinforcing steel, thereby destroying passive film and inducing the localized corrosion of the reinforcement [1,2,3]. The formation and accumulation of corrosion products lead to expansion stress and subsequently the cracking and spalling of concrete, further accelerating corrosion damage and reducing the mechanical property of the embedded reinforcement [4]. Therefore, the efficient corrosion protection of the reinforcement is of great importance for guaranteeing the durability of reinforced concrete in a marine environment.

Among different corrosion protection strategies, admixed organic corrosion inhibitors are regarded as an effective way to enhance the corrosion resistance of reinforced concrete [5,6,7]. Commonly used organic corrosion inhibitors are mainly based on alcohol amine, amine ester, fatty acid, etc. [8,9]; the inhibition mechanisms are closely related to chemical adsorption through the stable covalent bond by sharing lone-pair electrons or π electrons between polar groups (e.g., N, O and P), or heterocyclic groups for organic corrosion inhibitors, and vacant d orbitals of the reinforcement and physical adsorption through electrostatic attraction/Van der Waals force [10,11]. Based on the above chemical/physical adsorption, the organic corrosion inhibitors form a dense protection layer on the reinforcement surface, and thus efficiently prevent the adsorption and contact of O_2_ and chlorides, increasing the critical chloride concentration for corrosion initiation and reducing the corrosion damage rate of the reinforcement [8,9,12]. Based on XPS analysis, Jamil et al. [13] and Welle et al. [3] illustrated that chemical adsorption was relevant to dimethylethanolamine-based corrosion inhibitors, thus halting chloride adsorption and the corrosion damage of the reinforcing steel. Zhou et al. [14] found that organic corrosion inhibitors based on imidazole tetrafluoride salts were mainly physically adsorbed on the reinforcement surface through electrostatic attraction in chloride-containing simulated concrete pore solution (SPS), halting both the anodic and cathodic corrosion reaction of the immersed reinforcement.

The high inhibition efficiency of various admixed organic corrosion inhibitors has been widely reported in the literature. Shen et al. [15] investigated the inhibition effect of aminopropyltriethoxysilane corrosion inhibitor on the reinforcing steel embedded in concrete. After 1 month, the corrosion current density of the inhibitor-containing specimen was about 2 μA/cm^2^, which was significantly smaller than the control specimen (20 μA/cm^2^); the inhibition efficiency was about 90%. It was reported [16] that after immersion in SPS (containing 1 g/L chlorides) with an amino alcohol-based corrosion inhibitor, the open-circuit potential (OCP) of the reinforcing steel was kept at −376 mV; after 30 min of immersion, the corrosion inhibitor reduced the corrosion current density based on the potentio-dynamic polarization results, and inhibition efficiency was about 78%. Zhao et al. [17] found that the triethanolammonium corrosion inhibitor increased the charge transfer resistance of the reinforcing steel: after immersion in chloride-containing SPS for 16 h, the charge transfer resistance was 11 times higher for the inhibitor-containing sample compared to the inhibitor-free reference sample.

Based on the above literature review, it can be found that the interaction between the traditional admixed organic corrosion inhibitor and reinforcement and its corrosion-inhibition behavior has been strongly emphasized in previous studies. However, the influence of the traditional admixed organic corrosion inhibitor on the concrete matrix was less reported, especially regarding pore structure and the impermeability of concrete. This study belongs to a research project about the design and preparation of novel hydrophobic functional organic corrosion inhibitors for the corrosion protection of reinforced concrete serving in extremely harsh marine environments. The proposed organic corrosion inhibitor, on one hand, can strongly adsorb on the reinforcement surface and subsequently improve the corrosion resistance of the reinforcement; on the other hand, it can reduce the porosity and increase the hydrophobicity of concrete, halting chloride transport in the concrete matrix. The previous study [18] indicated that the prepared hydrophobic functional organic corrosion inhibitor effectively reduced the corrosion rate of the reinforcement in SPS (containing 0.1 mol/L chlorides): after 168 h of immersion, the polarization resistance (R_p_) of the reference sample was about 50.3 kΩ·cm^2^, and R_p_ increased to 77.1, 131.5, 192.4 and 213.6 kΩ·cm^2^ with 0.2, 0.4, 0.6 and 0.8 wt.% inhibitor in SPS, respectively. However, it is essential to clarify the influence of the hydrophobic functional organic corrosion inhibitor on the micro/macro properties of the concrete matrix and the corrosion performance of the reinforcement embedded in the concrete. To this end, the present study aims to evaluate the inhibition effect of the novel hydrophobic functional organic corrosion inhibitor on reinforced concrete. On one hand, the influence of the hydrophobic functional organic corrosion inhibitor on the micro/macro properties of concrete was investigated, including contact angle, pore structure and the water absorption behavior of concrete. On the other hand, the electrochemical behavior of the reinforcing steel embedded in concrete was characterized by OCP, electrochemical impedance spectroscopy (EIS) and potentio-dynamic polarization (PD).

## 2. Materials and Methods

### 2.1. Materials

The hydrophobic functional organic corrosion inhibitor (with a solid content of 48 wt.%, provided by Jiangsu Research Institute of Building Science, China) used in this study mainly consisted of glycosidic ketone, which was esterized and amidated from polyhydroxy amino organic molecules. As a result, the prepared hydrophobic functional organic corrosion inhibitor processes both hydrophobic groups and multisite adsorption groups, as shown in Figure 1. The designed inhibition mechanisms are illustrated as follows: when admixed in concrete with high alkalinity, the hydrophobic functional organic corrosion inhibitor is hydrolyzed with OH^−^ in concrete pore solution, releasing both a multisite adsorption group and a hydrophobic group. The released hydrophobic group can react with Ca^2+^ in concrete pore solution to form a hydrophobic film on pore walls, leading to the reduced porosity and increased hydrophobicity of concrete. In this way, chloride penetration in concrete is halted. Furthermore, the released multisite adsorption groups are efficiently adsorbed and generate a protection film on the steel surface, halting the adsorption of aggressive ions and enhancing the corrosion resistance of the reinforcing steel.

The cement used in this study was P.O. 42.5 ordinary Portland cement; its chemical compositions and particle size distribution are shown in Table 1 and Figure 2. The used fine aggregates were natural river sands with the physical properties and gradation are summarized in Table 2 and Table 3; the used coarse aggregates were crushed granite stones with the physical properties and gradation as shown in Table 4 and Table 5. The used reinforcement in this study was HPB235 carbon steel with a diameter of 8 mm; its chemical compositions are summarized in Table 6. Before measurements were taken, the reinforcement surface was polished by sandpapers from #400 to #2000. The polished reinforcement was cleaned with alcohol and stored in a vacuum-drying chamber. The used chemical reagents (Ca(OH)_2_, NaOH, KOH and NaCl) were analytically pure.

### 2.2. Sample Preparations

According to the literature [19], the recipe of simulated concrete pore solution (SPS) is as follows: saturated Ca(OH)_2_ + 0.06 mol/L NaOH + 0.180 mol/L KOH; pH of the prepared SPS is 12.7.

Table 7 shows the concrete mixture proportion in this study. The water-to-cement (w/c) ratio of cement paste and concrete is 0.5. The dimensions of cement paste and concrete were 4 cm × 4 cm × 16 cm and 10 cm × 10 cm × 10 cm, respectively. For the reinforced concrete specimen, the HPB235 carbon steel bar was centrally embedded in the specimen (the schematic diagram of reinforced concrete specimen in this study is shown in Figure 3). For concrete casting, cement and aggregates were first dry-mixed with a concrete mixer and liquid hydrophobic functional organic corrosion inhibitors with different proportions were dispersed into tape water. The mixed aqueous solution was then poured into the above dry blend material and continuously stirred 3 min, followed by casting the fresh concrete into the mold. For reinforced concrete, the top and bottom of the reinforcement were isolated with epoxy coating. The reinforcement was placed 2 cm from the edge of the concrete specimens and the working length and surface area of the reinforcement electrode were 8 cm and 20.096 cm^2^. The concentration of hydrophobic functional organic corrosion inhibitor in both cement paste and concrete was 0 wt.% (reference), 1 wt.%, 2 wt.% and 4 wt.% per dry cement weight, respectively. The curing conditions for the cement paste, concrete and reinforced concrete samples were 20 °C and 95% RH.

### 2.3. Methods

#### 2.3.1. Pore Structure Analysis

In this study, the effect of hydrophobic functional organic corrosion inhibitor on the pore structure of the cement paste was evaluated with mercury intrusion porosimetry (MIP). MIP is widely adopted for investigating the porosity and pore size distribution of cement-based materials, i.e., cement paste, mortar and concrete. Many factors affect the accuracy of MIP results, mainly including sampling, sample conditioning, pressure application rate, maximum applied intrusion pressure, used contact angle and the surface tension of mercury. In addition, the expansion of sample cells under pressure, differential mercury compression, sample compression and the hydrostatic head of mercury also affect the accuracy of the MIP results. The parameter used for MIP tests in this study is consistent with the reported studies to guarantee the accuracy of the derived results. After curing for 28 days, the cement paste sample was crushed into small pieces (smaller than 15 mm × 15 mm × 15 mm) and immersed in ethanol for 7 days to stop hydration; the sample was then dried at 40 °C in a vacuum-drying chamber for 14 days before the MIP tests. The pore structure of the cement paste sample was analyzed with Auto Pore 9500 IV (micromeritics).

Based on the MIP results, the pore diameter can be calculated based on the applied pressure using the Washburn equation [20]:D = −4γcosθ/P,(1)
where D is the pore diameter; γ is the surface tension of Hg (0.480 N/m); θ is the contact angle of Hg on the pore wall (130°); and P is the applied pressure (MPa). The applied pressure range was 0.0036–210 MPa with the determined pore diameter range of 5 nm to 350 μm.

#### 2.3.2. Contact Angle Measurement

The contact angle is used to evaluate the affinity between the solid surface and the liquid it is contacting: a smaller contact angle represents better surface wettability. A contact angle greater than 90° corresponds to a solid surface which is not wettable with the liquid. Otherwise, a contact angle less than 90° corresponds to a surface presenting good wettability with the formation of a larger solid–liquid interface.

The contact angle was determined for both the cement paste and the concrete samples containing the hydrophobic functional organic corrosion inhibitor with different concentrations. The measuring range of the used contact angle instrument was from 1° to 180° with a resolution of 0.01°. At 28 days’ curing age, the contact angles at at least 10 different locations were measured after natural drying; the aggregates were carefully avoided when measuring the contact angle of concrete samples. There were 3 replicates for each specimen.

#### 2.3.3. Capillary Water Absorption Test

The effect of the hydrophobic functional organic corrosion inhibitor on the capillary water absorption of both the cement paste and the concrete was evaluated according to the ASTM-C1585 standard. The advantage of the capillary water absorption test is that this measurement is quite easy to be conducted in a laboratory using the gravimetric technique. The shortcomings of the capillary water absorption test can be described as follows: first, the penetration depth and distribution of water molecules are unable to be obtained via this method. Furthermore, the influence of gravity on the capillary absorption coefficient should also be considered in long-term experiments. After curing for 28 days, the cement paste and concrete samples were dried at 40 °C until they reached a constant weight in the vacuum-drying chamber. The specimen weight was measured at an interval of 24 h until the fully dried state was achieved (about 7 days), i.e., the weight loss of the sample within 24 h was lower than 0.02 %. Afterwards, the other 5 surfaces were isolated with epoxy and only one surface was exposed to water the during capillary water absorption measurement to ensure one-way water transport [21]. The top surface was loosely covered with a piece of plastic film to avoid water evaporation. The exposed surface was immersed in water of about 5 mm depth and the weight change in the sample was recorded at different time intervals. The weight difference before and after water absorption is the cumulative water absorption quality of the sample at a specific time interval. For each measurement, the specimen was quickly taken out, wiped with a dry towel to remove the free water on the surface, and then placed on the balance with the wet side up. The specimen was put back in water immediately to continue the water absorption test after weighing. The weighing process was completed within 15 s. The capillary water absorption amount per surface area at different time intervals can be calculated using the following equation [21]:Δw(t) = (w_t_−w_0_)/A(2)
where ΔW(t) is the capillary water absorption amount per surface area at a specific time interval (g/m^2^); w_0_ is the initial sample weight before water absorption (g); w_t_ is the sample weight after water absorption at a specific time interval (g); and A is the exposed surface area of the cement paste or concrete sample (m^2^).

Subsequently, the capillary water absorption coefficient of the cement or concrete sample can be calculated as follows:k = Δw(t)·t^−1/2^(3)
where k is the capillary water absorption coefficient (kg·m^−2^·h^−1/2^) and t is the absorption time (h).

#### 2.3.4. Characterization of Adsorption Amount of Hydrophobic Functional Organic Corrosion Inhibitor

In this study, the adsorption amount of the hydrophobic functional organic corrosion inhibitor on the surface of HPB235 carbon steel in SPS was evaluated using total organic carbon amount (TOC). The initial TOC of the SPS containing the hydrophobic functional organic corrosion inhibitor with different concentrations (1, 2 and 4 wt.%) was determined using a total organic carbon analyzer (TOC-L, Shimadzu). Afterwards, HPB235 carbon steel with the dimensions of 40 mm × 40 mm × 5 mm and a working surface area of 28.660 cm^2^ was immersed in the above relevant SPS. At the immersion age of 7, 14 and 28 days, the TOC of the collected top SPS was measured. Based on the difference in the TOC of the SPS before and after adsorption, the adsorption amount of the hydrophobic functional organic corrosion inhibitor on the carbon steel surface of the SPS was calculated.

#### 2.3.5. Electrochemical Behavior of Reinforced Concrete

After 28 days of curing, reinforced concrete samples were half-immersed in 3.5 wt.% NaCl solution. The influence of the hydrophobic functional organic corrosion inhibitor on the electrochemical behavior of the reinforced concrete was evaluated using OCP, EIS and PD measurements at different immersion time intervals (1, 7, 14, 28, 56 and 72 days). The used equipment was Metrohm Autolab-Potentionstat PGSAT 302N, and a three-electrode set-up containing a reference electrode (saturated calomel electrode), a working electrode (reinforcement electrode) and a counter electrode (Ti mesh) was used for electrochemical measurements at 25 °C ± 1 °C. EIS measurements were conducted in 10^−2^ Hz–10^−5^ Hz at 10 mV. PD measurement was conducted in −200 mV–1000 mV (vs. OCP) with 0.5 mV·s^−1^.

## 3. Results and Discussions

### 3.1. Effect of Hydrophobic Functional Organic Corrosion Inhibitor on Pore Structure of Cement Paste

The total porosity and pore size distribution of the cement paste with different dosages of the hydrophobic functional corrosion inhibitor at 28 days of curing are presented in Figure 4. It is shown in Figure 4a that no obvious peak was observed when the pore diameter was larger than 150 nm, indicating that all pores in the cement paste were less than 150 nm. In addition, a higher dosage of the corrosion inhibitor was related to a lower cumulative mercury intrusion. The total porosity of the reference sample was about 0.161 mL/g and reduced to 0.153 mL/g with 1 wt.% dosage of the hydrophobic functional corrosion inhibitor (Figure 4a). When the dosage of the corrosion inhibitor increased to 2 and 4 wt.%, the total porosity was further reduced to 0.142 and 0.139 mL/g, respectively. This indicated that the pore structure of the cement paste was refined by the hydrophobic functional corrosion inhibitor. As shown in Figure 4b, a similar pore size distribution was found in both the reference sample and the inhibitor-containing sample. However, the peak corresponding to large pores (at about 100 nm) shifted to a smaller pore size, and the corresponding peak intensity was also reduced by the hydrophobic functional corrosion inhibitor. The critical pore diameter was 77.1 nm, 69.0 nm, 55.8 nm and 40.0 nm for the sample with 0, 1, 2 and 4 wt.% dosages of hydrophobic functional corrosion inhibitor, respectively.

The pore volumes and the different pore size ranges for the cement paste samples at 28 d of curing are summarized in Table 8. It was observed that, compared to the reference sample, the pore volume in the pore size range of 50–100 nm and the pore size larger than 100 nm was reduced by the hydrophobic functional corrosion inhibitor; meanwhile, the pore volume in the pore size range of 20–50 nm and the pore sizes smaller than 20 nm increased. This confirms that the hydrophobic functional corrosion inhibitor effectively reduced the content of large pores and increased the content of small pores, leading to a refined pore structure of the cement paste.

### 3.2. Effect of Hydrophobic Functional Corrosion Inhibitor on Contact Angle of Cement-Based Materials

Figure 5 presents the contact angle results for both cement paste and concrete at 28 d of curing. It can be observed in Figure 5a that the water droplet well wetted the surface of the cement paste for the reference sample. The contact angle for the reference sample was in the range of 4.6° to 40°, with the average value of 19.7°, as shown in Figure 5b. After admixing with the hydrophobic functional corrosion inhibitor, the wettability of the water droplets was significantly reduced and the water droplets maintained partial sphere morphology on the cement paste surface. As a result, when the concentration of the hydrophobic functional corrosion inhibitor was 1 wt.%, the contact angle increased to about 32.2–103° with the higher average value of 59.4°. When the concentration of the hydrophobic functional corrosion inhibitor increased to 2 and 4 wt.%, the contact angle further increased to about 58.9–110.1° with the average value of 80.5°.

Compared to the cement paste, the effect of the hydrophobic functional corrosion inhibitor on the contact angle was less pronounced for the concrete sample, as shown in Figure 5c,d. This might be related to the more heterogeneous characteristics of the concrete sample due to the presence of fine and coarse aggregates. However, compared to the reference sample, the higher contact angle was still relevant to the samples in the presence of the hydrophobic functional corrosion inhibitor. For example, when the 4 wt.% hydrophobic functional corrosion inhibitor was used, the contact angle was in the range of about 20.0–56.8° with an average value of 38.9°, which was significantly higher than the reference sample (with the contact angle of almost 0°). The above results indicate that the hydrophobic functional corrosion inhibitor effectively increased the water contact angle on the cement paste surface and the beneficial effect was enhanced by the higher inhibitor dosage. The increased contact angle combined with the refined pore structure (confirmed in Section 3.1) might result in halted water and chloride transport, which is beneficial for improving the corrosion resistance of reinforced concrete.

### 3.3. Effect of Hydrophobic Functional Corrosion Inhibitor on Capillary Water Absorption Amount of Cement-Based Materials

The capillary water absorption amounts for both cement paste and concrete with different dosages of hydrophobic functional corrosion inhibitor are presented in Figure 6. For cement paste (Figure 6a), it was observed that the reference sample exhibited the largest capillary water absorption amount of 7.67 kg·m^−2^ after 48 h of immersion. In the presence of the hydrophobic functional corrosion inhibitor, the capillary water absorption amount in the cement paste was significantly reduced; the lower capillary water absorption amount was relevant to the larger dosage of the hydrophobic functional corrosion inhibitor in the cement paste. After 48 h of immersion, the capillary water absorption amounts of cement paste were 5.81, 4.81 and 3.94 kg·m^−2^ with the inhibitor dosages of 1, 2 and 4 wt.%, which were reduced by 24.3%, 37.3% and 48.6% compared to the reference sample. The capillary water absorption amounts of concrete were all smaller compared to cement paste. However, a similar trend was also observed for the concrete samples (Figure 6b): After 72 h of immersion, the capillary water absorption amounts of concrete with 1, 2 and 4 wt.% hydrophobic functional corrosion inhibitor were 1.26, 0.97 and 0.92 kg·m^−2^, which were reduced by 54.2.8%, 64.7% and 66.5% compared with the reference sample. Based on the contact angle results (Figure 5), the hydrophobic functional corrosion inhibitor enhanced the hydrophobic properties of the cement/concrete sample, and this positive effect was more pronounced with higher inhibitor concentrations. Therefore, the cement paste and concrete samples with 4 wt.% inhibitor presented lower capillarity than samples with 2 wt.% inhibitor, even though both samples presented similar total porosity (Figure 4).

Based on Equation (3), the capillary water absorption coefficients (k) for cement paste and concrete with different dosages of hydrophobic functional corrosion inhibitor were calculated, as presented in Figure 7. For both cement paste and concrete, the reference sample presented a very high initial capillary absorption coefficient which was gradually reduced with immersion time. In the presence of the hydrophobic functional corrosion inhibitor, the initial capillary water absorption coefficient was significantly reduced, and the lower initial capillary water absorption coefficient was relevant to the concrete with higher dosages of corrosion inhibitor. As a result, the reduction in the capillary water absorption coefficient of the concrete with the hydrophobic functional corrosion inhibitor was not as obvious as the reference sample. However, the capillary water absorption coefficients of both the cement paste and the concrete with the hydrophobic functional corrosion inhibitor were still significantly lower than the reference sample. The above results indicate that the capillary water absorption of the cement-based materials was efficiently delayed by the hydrophobic functional corrosion inhibitor.

### 3.4. Adsorption Amount of Hydrophobic Functional Organic Corrosion Inhibitor

Figure 8 shows the adsorption amount of hydrophobic functional organic corrosion inhibitor on the surface of the HPB235 carbon steel in the SPS after different immersion ages of 1, 7, 14, and 28 days. It was observed that the hydrophobic functional organic corrosion inhibitor was mainly adsorbed on the carbon steel surface during the first 7 days. Afterwards, the adsorption rate was gradually reduced. For example, the adsorption amount on the reinforcement surface was about 273.18 mg/(L·cm^2^) in the SPS with 4 wt.% hydrophobic functional organic corrosion inhibitor after 7 d of immersion, which was already 90% of the adsorption amount after 28 d of immersion in the same solution (303.35 mg(L·cm^2^)). Furthermore, the larger adsorption amount was relevant to the larger initial content of hydrophobic functional organic corrosion inhibitor in SPS. After 28 d of immersion, the adsorption amount was 109.11, 149.43 and 303.35 mg(L·cm^2^) for the samples immersed in SPS with 1, 2 and 4 wt.% hydrophobic functional organic corrosion inhibitor, respectively. The results indicated that effective adsorption was relevant to the hydrophobic functional organic corrosion inhibitor on the carbon steel surface; the adsorbed corrosion inhibitor might increase the corrosion resistance of reinforced concrete, which was confirmed by the electrochemical measurements, as described in detail further below.

### 3.5. Effect of Hydrophobic Functional Corrosion Inhibitor on Electrochemical Behavior of Reinforced Concrete

#### 3.5.1. OCP of the Reinforcement Embedded in Concrete

The OCP of the reinforcement in the concrete immersed in 3.5 wt.% NaCl solution is presented in Figure 9. A very negative OCP of the embedded reinforcement (lower than −600 mV) was observed for the reference sample during the first 19 d of immersion. Afterwards, the OCP of the reinforcement was positively shifted and maintained at about −400 mV after 72 d of immersion. The OCP of the reinforcement embedded in concrete admixed with the hydrophobic functional organic corrosion inhibitor was significantly more positive, compared with the reference sample. At 72 days, the OCP of the reinforcement in the concrete with 1, 2 and 4 wt.% hydrophobic functional organic corrosion inhibitor was −80 mV, −244 mV and −176 mV, respectively.

Due to the difficult contact between O_2_ and the reinforcement, normally the reinforcement embedded in concrete with a low internal humidity exhibits a more positive OCP [22]. In the present study, the admixed hydrophobic functional organic corrosion inhibitor both reduced the total porosity (Figure 4) and increased the water contact angle (Figure 5) of the cement-based materials. As a result, water transport was significantly halted in the concrete admixed with the hydrophobic functional organic corrosion inhibitor (Figure 6 and Figure 7), thus leading to a reduced internal humidity of the concrete and, subsequently, the more positive OCP of the embedded reinforcement noted above. It was also reported [23] that the OCP was closely related to the possibility of corrosion damage of the reinforcement: when the OCP was higher than −120 mV, the possibility for corrosion damage was only 10%; when the OCP was more negative than −270 mV, the possibility for corrosion damage was 90%. Therefore, the positive OCP shift of the reinforcement embedded in the concrete admixed with the hydrophobic functional organic corrosion inhibitor indicated that the hydrophobic functional organic corrosion inhibitor reduced the possibility of corrosion damage of the reinforcement.

#### 3.5.2. EIS of the Reinforcement Embedded in Concrete

Figure 10 presents EIS of the reinforcement embedded in concrete immersed in 3.5 wt.% NaCl solution. It was observed that the impedance |Z| of all samples increased during the first 56 d of immersion; at 72 d immersion, the |Z| of the reinforcement was slightly reduced. However, the impedance |Z| and phase angle of the reinforcement embedded in the concrete admixed with the hydrophobic functional organic corrosion inhibitor were both significantly higher than the reference sample at different immersion ages. The beneficial effect of the hydrophobic functional organic corrosion inhibitor was more pronounced with a higher dosage in the concrete: when 1 and 2 wt.% hydrophobic functional organic corrosion inhibitor were used, the reinforcement exhibited a similar |Z| (e.g., 2489–2889 Ω at 7 d of immersion) and phase angle (e.g., about 60° at 7 d of immersion); however, when the dosage of the hydrophobic functional organic corrosion inhibitor increased to 4 wt.%, the |Z| (e.g., 5683 Ω at 7 d immersion age) and phase angle (e.g., about 70° at 7 d of immersion) of the embedded reinforcement were further increased.

In this study, a Randles circuit (R_s_(R_ct_Q_dl_), embedded in Figure 10) was applied for the EIS fitting of the reinforcement embedded in concrete. In the relevant circuit, R_s_ is concrete resistance, including electrolyte resistance; R_ct_ is the charge transfer resistance of the embedded reinforcement; and Q_dl_ represents the characteristic of the electrical double layer. A constant phase element (CPE) was used to represent the heterogeneity at the reinforcement/concrete interface [24,25,26]. The fitted R_ct_ and Q_dl_ of the reinforcement embedded in the concrete are summarized in Figure 11. The R_ct_ of all samples increased during the first 56 d of immersion; after 72 d of immersion, the R_ct_ of the reinforcing steel was slightly reduced. At each immersion age, the R_ct_ of the reinforcement embedded in concrete admixed with the hydrophobic functional organic corrosion inhibitor was one order of magnitude higher than the reference sample; correspondingly, the Q_dl_ of the reinforcement embedded in the concrete admixed with the hydrophobic functional organic corrosion inhibitor was lower than the reference sample. Furthermore, the higher R_ct_ of the embedded reinforcement was relevant to the specimen with a higher dosage of the hydrophobic functional organic corrosion inhibitor. For example, at 56 d of immersion, the R_ct_ values of the reinforcement embedded in concrete with 1, 2 and 4 wt.% hydrophobic functional organic corrosion inhibitor were 6.071, 8.310 and 25.578 kΩ/cm^2^, which were significantly higher than the R_ct_ values of the reference sample (0.330 kΩ/cm^2^). The above experimental results indicate that the hydrophobic functional organic corrosion inhibitor effectively improved the corrosion resistance of the reinforced concrete.

#### 3.5.3. PD Results of the Reinforcement Embedded in Concrete

Figure 12 presents the PD curves of the reinforcement embedded in concrete. At all immersion ages, the corrosion potential of the reinforcement in concrete admixed with the hydrophobic functional organic corrosion inhibitor was more positive, compared with the reference sample. Furthermore, in the presence of the hydrophobic functional organic corrosion inhibitor, the anodic current density (corresponding to the dissolution of the embedded reinforcement) was dramatically reduced; however, the alteration in cathodic current density (corresponding to oxygen reduction) was not very obvious. Therefore, the hydrophobic functional organic corrosion inhibitor acted as an anodic corrosion inhibitor, mainly reducing the dissolution rate of the embedded reinforcement in concrete.

Figure 13 presents the calculated corrosion current density of the embedded reinforcement and the inhibition efficiency of the hydrophobic functional organic corrosion inhibitor based on the derived PD curves. Generally, the corrosion current density of the reinforcement embedded in concrete admixed with the hydrophobic functional organic corrosion inhibitor (in the range of 5.39 × 10^−8^−4.37 × 10^−7^ A/cm^2^) was one order of magnitude lower than the reference sample (in the range of 1.29 × 10^−6^−3.47 × 10^−6^ A/cm^2^). The significantly reduced corrosion current density was consistent with the increased R_ct_ of the reinforcement derived from the EIS results in Section 3.5.2. Due to the excellent inhibition effect, the inhibition efficiency of the hydrophobic functional organic corrosion inhibitor was higher than 80% at most immersion ages; a higher dosage of corrosion inhibitor maintained the high inhibition efficiency for a longer period.

#### 3.5.4. Discussions

Based on the experimental results in this study, the hydrophobic functional organic corrosion inhibitor significantly altered the performance of the concrete and embedded reinforcement. For the concrete, the pronounced pore refinement was relevant to the specimen in the presence of the hydrophobic functional organic corrosion inhibitor, as evidenced by the reduced total porosity and content of large pores (Figure 4 and Table 8). Furthermore, the average water contact angle was increased to 59.4–80.5° for cement paste and 16.3–38.9° for concrete in the presence of the hydrophobic functional organic corrosion inhibitor (Figure 5), indicating that the hydrophobic property of the concrete was increased by the inhibitor. The refined pore structure and increased hydrophobic property reduced the capillary water absorption amount of the concrete (Figure 7), which was beneficial for halting the chloride transport in the concrete and the corrosion initiation of the reinforcement. Due to the lower internal humidity and chloride content caused by the halted water and chloride transport, the open-circuit potential of the reinforcement was more positive when embedded in concrete in the presence of hydrophobic functional organic corrosion inhibitor (Figure 9). Furthermore, the effective adsorption on the reinforcement surface was relevant to the hydrophobic functional organic corrosion inhibitor (Figure 8), mainly due to reducing the anodic corrosion reaction rate and significantly improving the corrosion resistance of the reinforced concrete, as evidenced by the charge transfer resistance that was one order of magnitude higher and the corrosion current density of the embedded reinforcement that was one order of magnitude lower (Figure 11 and Figure 13). As a result, the novel hydrophobic functional organic corrosion inhibitor exhibited very high inhibition efficiency (in the range of 71–96%) and obviously the better corrosion-inhibition effect was relevant to the higher dosage of corrosion inhibitor in the concrete (96% in the presence of 4 wt.% inhibitor).

Table 9 shows a comparison of the inhibition efficiency of the hydrophobic functional organic corrosion inhibitor and the conventional corrosion inhibitors found in the reported studies on concrete [27,28,29,30,31,32,33,34]. It can be observed in Table 9 that, due to the different compositions and concentrations of corrosion inhibitors, together with the different concentrations of chlorides in concrete, the reported inhibition efficiencies in previous studies were of a very wide range. However, the hydrophobic functional organic corrosion inhibitor proposed in the present study exhibited a very high inhibition efficiency in concrete, compared with the other corrosion inhibitors in the reported studies. Meanwhile, the excellent inhibition effect was maintained for a very long time.

## 4. Conclusions

In this study, the corrosion performance of reinforced concrete in the presence of a novel hydrophobic functional organic corrosion inhibitor was evaluated. The hydrophobic functional organic corrosion inhibitor reduced the total porosity and amount of large pores, leading to a refined pore structure in the cement paste. Furthermore, the capillary water absorption amount of the cement-based materials was significantly reduced due to the reduced porosity and enhanced hydrophobic properties of the cement-based materials due to the hydrophobic functional organic corrosion inhibitor.

The embedded reinforcement exhibited more positive open-circuit potential and a higher corrosion resistance in concrete admixed with the hydrophobic functional organic corrosion inhibitor due to the halted water and chloride transport process and, subsequently, the lower internal humidity of the concrete. Furthermore, the hydrophobic functional organic corrosion inhibitor was effectively adsorbed on the reinforcing steel surface, improving the corrosion resistance of the embedded reinforcement. The better corrosion-inhibition effect was relevant to the higher dosage of corrosion inhibitor in concrete. Therefore, the novel hydrophobic functional organic corrosion inhibitor exhibited very high inhibition efficiency, and can be potentially used for the efficient corrosion protection of reinforced concrete in severe marine environments.

## Figures and Tables

**Figure 1 materials-15-07124-f001:**
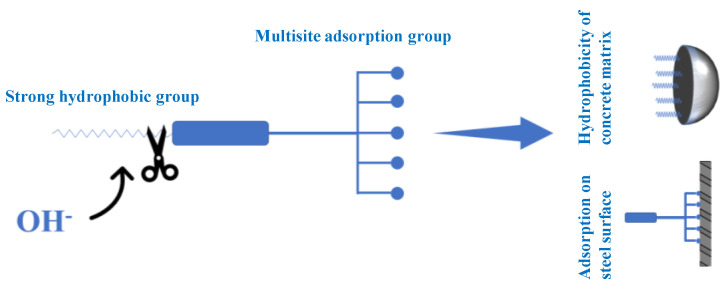
The schematic diagram of hydrophobic functional organic corrosion inhibitor.

**Figure 2 materials-15-07124-f002:**
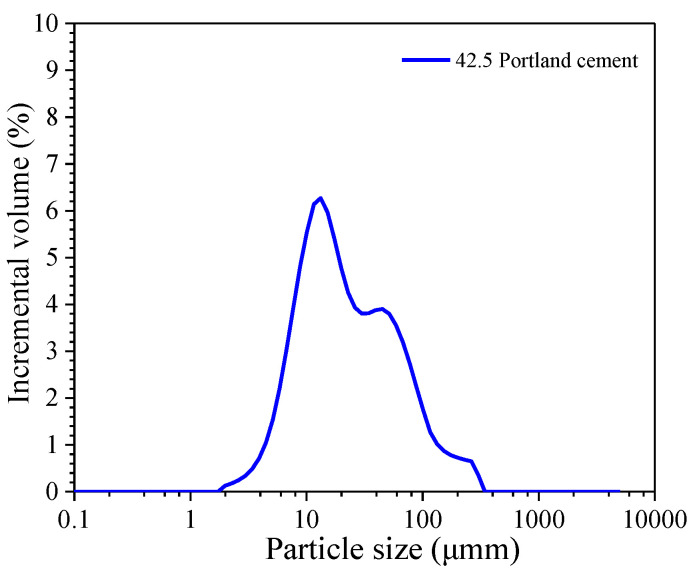
Particle size distribution of P.O. 42.5 ordinary Portland cement.

**Figure 3 materials-15-07124-f003:**
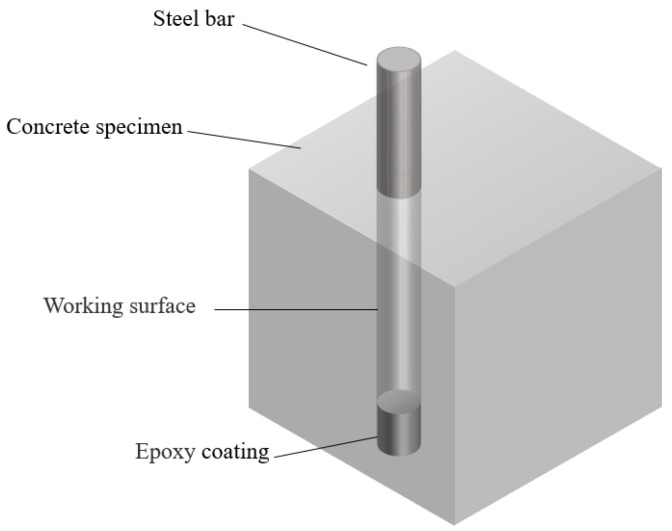
Schematic diagram of reinforced concrete specimens.

**Figure 4 materials-15-07124-f004:**
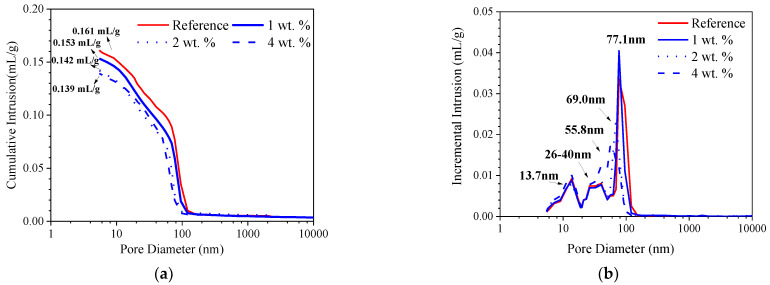
Total porosity (**a**) and pore size distribution (**b**) of cement paste with different dosages of hydrophobic functional corrosion inhibitor at 28 days of curing.

**Figure 5 materials-15-07124-f005:**
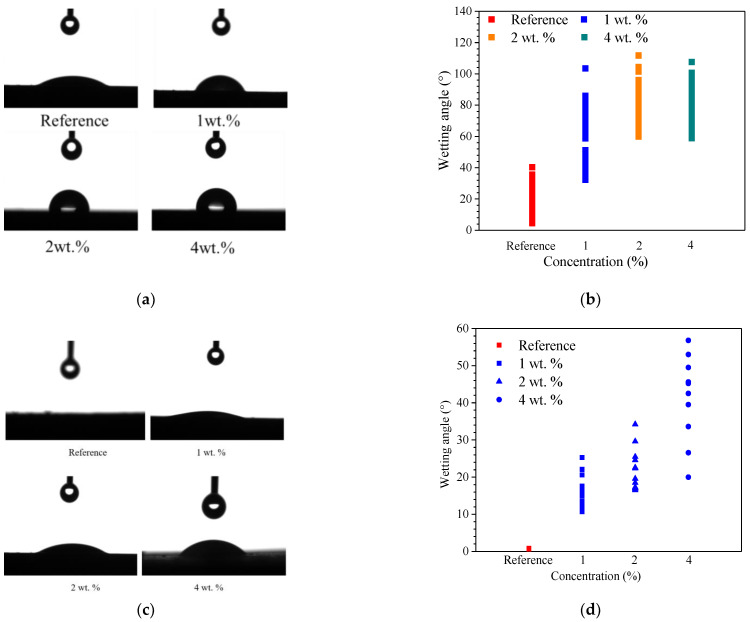
The contact angle results for both cement paste and concrete with different dosages of hydrophobic functional corrosion inhibitor at 28 d of curing. (**a**) Image of water droplets on cement paste sample; (**b**) contact angle; (**c**) image of water droplets on concrete samples; (**d**) contact angle.

**Figure 6 materials-15-07124-f006:**
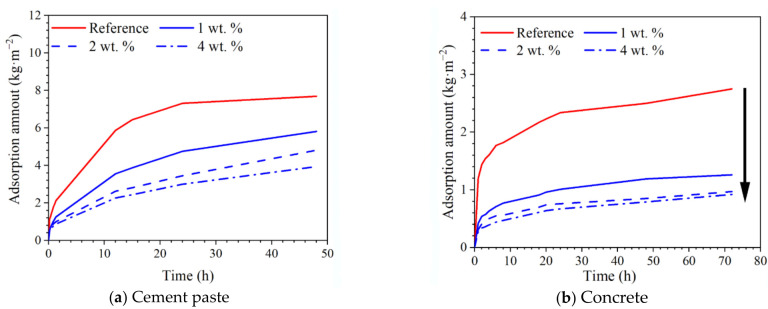
Capillary water absorption amount of both cement paste and concrete with different dosages of hydrophobic functional corrosion inhibitor at 28 d of curing.

**Figure 7 materials-15-07124-f007:**
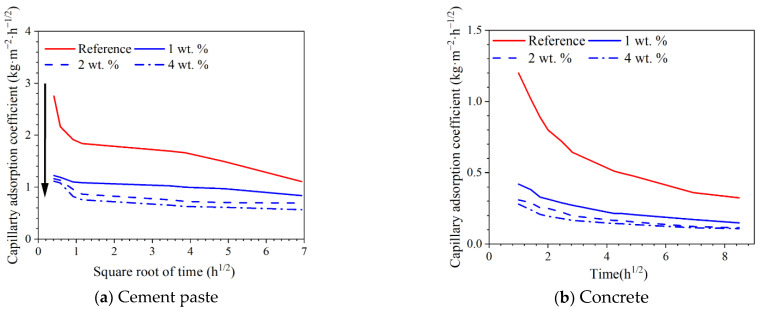
Capillary water absorption coefficient of both cement paste and concrete with different dosages of hydrophobic functional corrosion inhibitor at 28 d of curing.

**Figure 8 materials-15-07124-f008:**
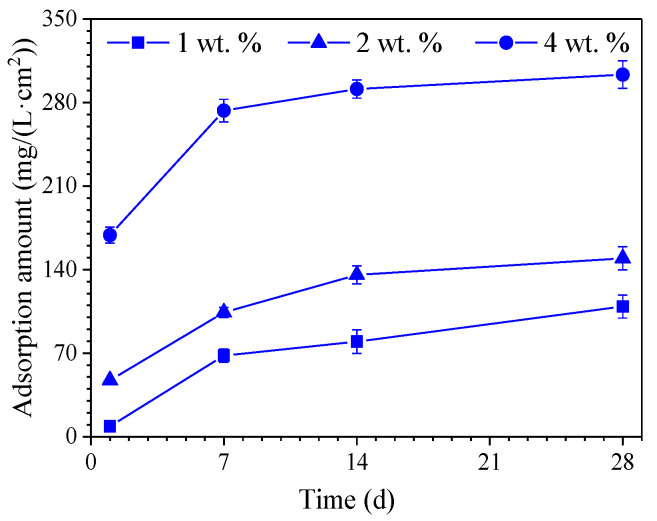
The adsorption amount of hydrophobic functional organic corrosion inhibitor on the surface of HPB235 carbon steel in SPS at different immersion ages.

**Figure 9 materials-15-07124-f009:**
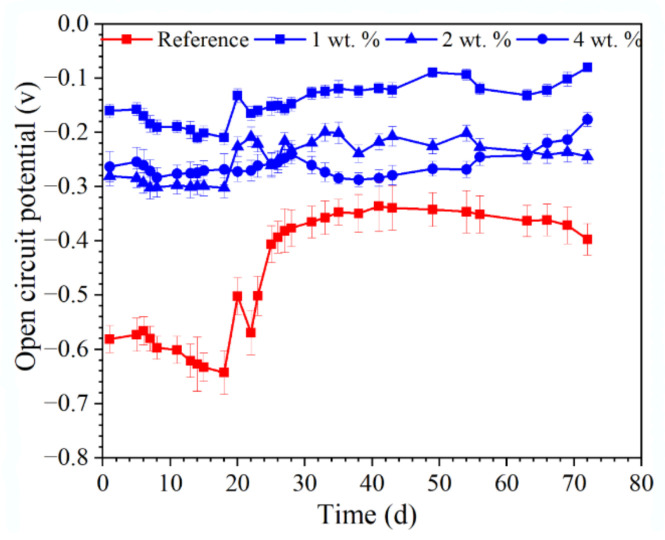
OCP of the reinforcement imbedded in concrete immersed in 3.5 wt.% NaCl solution.

**Figure 10 materials-15-07124-f010:**
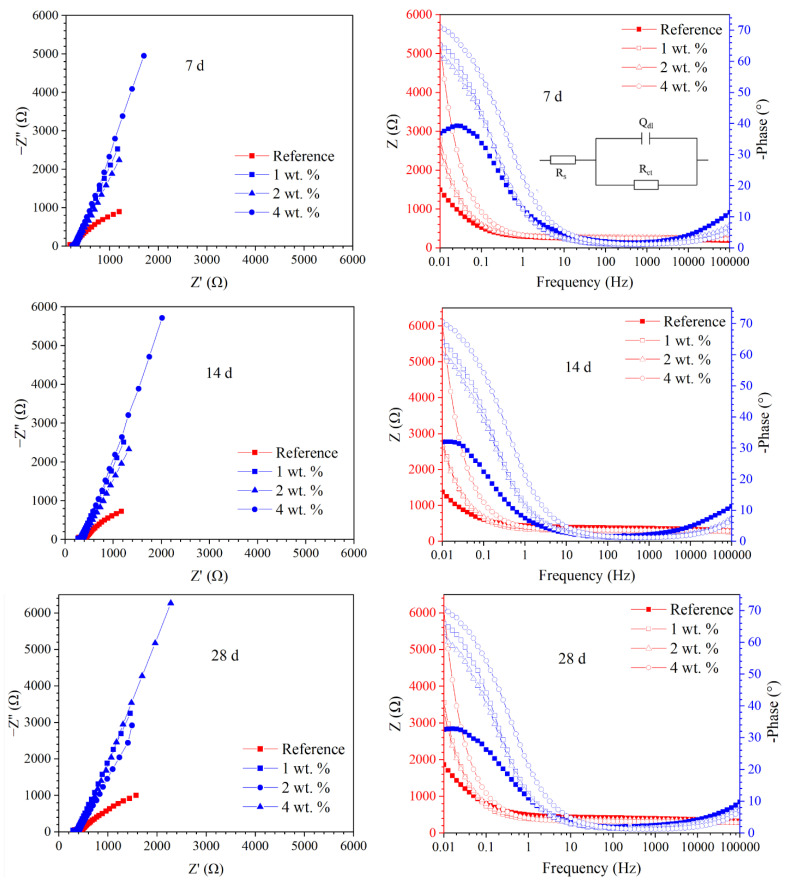
EIS of the reinforcement embedded in concrete immersed in 3.5 wt.% NaCl solution at different ages in both Nyquist and Bode format.

**Figure 11 materials-15-07124-f011:**
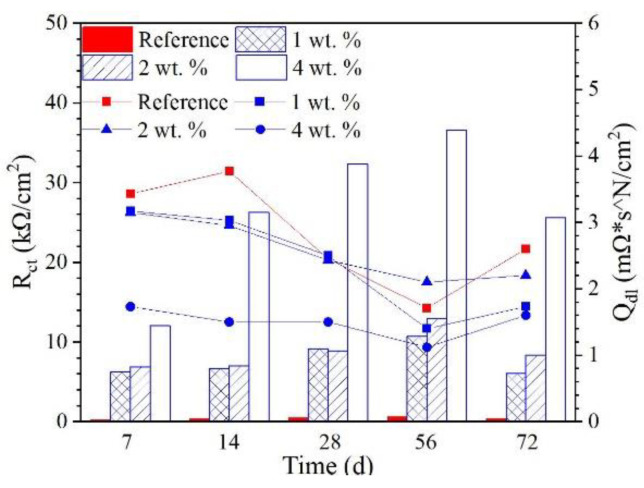
The fitted R_ct_ and Q_dl_ of the reinforcement embedded in concrete immersed in 3.5 wt.% NaCl solution at different ages.

**Figure 12 materials-15-07124-f012:**
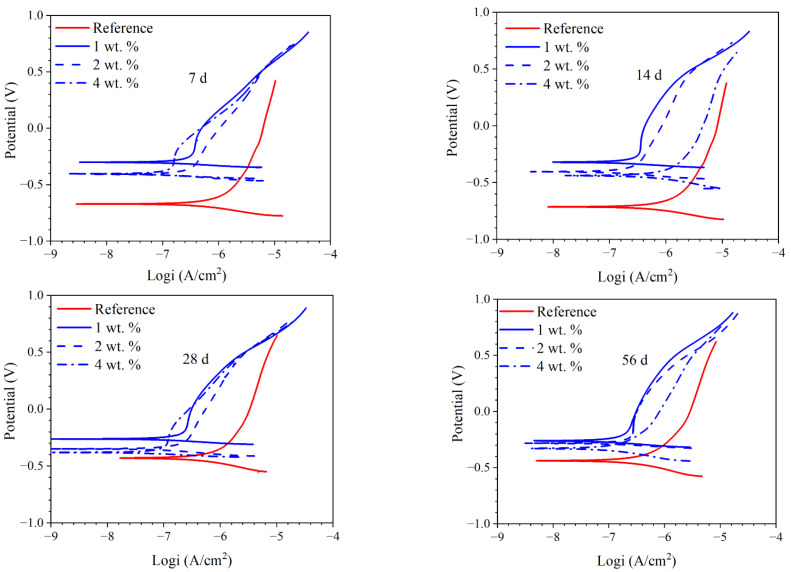
PD curves of the reinforcement embedded in concrete immersed in 3.5 wt.% NaCl solution at different ages (7 d, 14 d, 28 d, 56 d and 72 d).

**Figure 13 materials-15-07124-f013:**
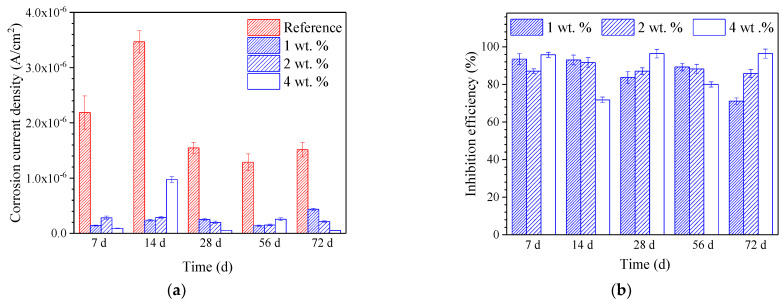
Calculated corrosion current density of the embedded reinforcement and inhibition. (**a**) Corrosion current density; (**b**) inhibition efficiency.

**Table 1 materials-15-07124-t001:** Chemical compositions of P.O. 42.5 ordinary Portland cement (wt.%).

Composition	SiO_2_	CaO	Al_2_O_3_	Fe_2_O_3_	SO_3_	MgO	K_2_O	Na_2_O	Others	LOI *
**Content**	21.51	58.05	7.37	4.16	2.85	1.76	0.51	0.16	1.71	1.92

* LOI: loss on ignition.

**Table 2 materials-15-07124-t002:** The physical properties of fine aggregates used in this study.

Mud Content (%)	Apparent Density (kg/m^3^)	Loose Bulk Density (kg/m^3^)	Close Packing Density (kg/m^3^)	Air Void (%)	Fineness Modulus
1.5	2610	1490	1557	40	2.6

**Table 3 materials-15-07124-t003:** The gradation of fine aggregates used in this study.

**Screen Size (mm)**	10.000	5.000	2.500	1.250	0.630	0.315	0.160
**Total residue (%)**	0.0	0.5	10.9	32.7	55.8	74.7	88.4

**Table 4 materials-15-07124-t004:** The physical properties of coarse aggregates used in this study.

Mud Content (%)	Apparent Density (kg/m^3^)	Loose Bulk Density (kg/m^3^)	Close Packing Density (kg/m^3^)	Air Void (%)
0.3	2733	1405	1518	9.1

**Table 5 materials-15-07124-t005:** The gradation of coarse aggregates used in this study.

**Screen Size (mm)**	16.000	10.000	5.000	2.5000
**Total residue (%)**	0	7.4	98.7	100

**Table 6 materials-15-07124-t006:** Chemical compositions of the steel reinforcement in this study (wt.%).

**Element**	C	Si	Mn	S	P	Fe
**Content**	0.15	0.29	0.56	0.04	0.03	98.93

**Table 7 materials-15-07124-t007:** Mixture proportions of concrete.

	Cement (kg/m^3^)	Fine Aggregates (kg/m^3^)	Coarse Aggregates (kg/m^3^)	Water (kg/m^3^)	Inhibitor (kg/m^3^)
CEM-0	340	1050	780	170	0
CEM-1	340	1050	780	166.6	3.4
CEM-2	340	1050	780	163.2	6.8
CEM-4	340	1050	780	156.4	13.6

**Table 8 materials-15-07124-t008:** The pore volume at different pore size ranges for cement paste in the presence of hydrophobic functional corrosion inhibitor with different concentrations at 28 d of curing.

	<20 nm (%)	20 nm–50 nm (%)	50 nm–100 nm (%)	>100 nm (%)
Reference	5.05	4.11	11.39	4.51
1 wt.%	5.62	4.85	11.63	2.63
2 wt.%	5.27	5.32	10.93	1.46
4 wt.%	5.30	6.16	9.86	1.16

**Table 9 materials-15-07124-t009:** Comparison of the inhibition efficiency of the hydrophobic functional organic corrosion inhibitor in this study and corrosion inhibitors in the reported studies [27,28,29,30,31,32,33,34].

Type of Corrosion Inhibitor	NaCl	Inhibitor	Time/d	Efficiency/%
Reed leaves extract	3.5 wt.%	0.5 wt.%	180	77.0
Ginger extract	8 wt.%	4 wt.%	60	40.9
Ca(NO_2_)_2_	8 wt.%	4 wt.%	60	25.7
Kelp extract	8 wt.%	4 wt.%	60	52.8
EG/AgNPs	Natural seawater	5 wt.%	90	82.6
Rice husk ash	5 wt.%	20 wt.%	40	85.0
Triethanolamine	3.5 wt.%	1 wt.%	28	64.2
Triethanolamine + Ca(NO_2_)_2_	3.5 wt.%	1 wt.%	28	90.1
Guanidine	1 M	0.5 M	30	77.5
1,6-Hexamethylenediamine	1 M	1 M	30	84.5
3-Aminopropyltriethoxysilane	1 M	1 M	30	62.6
Ethanolamine	3.5 wt.%	5 L/m^3^	270	95.0
Monoflurophosphate	3.5%	2%	270	50.0
This study	3.5%	4%	72	96.0

## Data Availability

Not applicable.

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
