# Peer review of "Inhibition Effect of Hydrophobic Functional Organic Corrosion Inhibitor in Reinforced Concrete"

_materials, 2022, doi:10.3390/ma15207124_

Round 1

Reviewer 1 Report

- The presented introduction is pretty modest.  Please include the latest research studies related to your work preferably between 2018 and 2022.
- Please include a subsection in the introduction as a research significance to explain the novelty of your work.
- Please include a brief but critical review regarding the conducted research studies in the introduction.
- Please add more explanation on the proposed methodology, verification process, and limitations.
- Please add more detail on the reported outcomes in Figure 3. Please add statistical characteristics of the graph accordingly.
- Please discuss the shortcomings of the conducted capillary water absorption test.
- Please elaborate on the effect of adding more than 4% water content on the reported outcomes in Figure 8.
- Please add statistical analyses to further discuss the proposed method in the discussion. In addition, please add a comparative discussion on the most important parameters that can affect the presented reports in this study.
- Please revise the conclusion to present a condensed version highlighting the main contributions of your work.

Author Response

Reviewer 1

- The presented introduction is pretty modest.  Please include the latest research studies related to your work preferably between 2018 and 2022.

- Thanks for the comments. According to the reviewer’s comment, Introduction section was revised and the latest research studies related to our work preferably between 2018 and 2022 were added into Introduction section in the revised manuscript as follows: “Shen et al. [15] investigated the inhibition effect of aminopropyltriethoxysilane corrosion inhibitor on the reinforcing steel embedded in concrete. After 1 month, the corrosion current density of the inhibitor-containing specimen was about 2 μA/cm2, which was significantly smaller than control specimen (20 μA/cm2); the inhibition efficiency was about 90 %. It was reported [16] that after immersed in SPS (containing 1 g/l chlorides) with amino alcohol based corrosion inhibitor, open circuit potential (OCP) of the reinforcing steel was kept at -376 mV; after 30 min immersion, the corrosion inhibitor reduced corrosion current density based on potentio-dynamic polarization results and its inhibition efficiency was about 78 %. Zhao et al. [17] found that triethanolammonium corrosion inhibitor increased the charge transfer resistance of the reinforcing steel: after immersed in chloride-containing SPS for 16 h, the charge transfer resistance was 11 times higher for inhibitor-containing sample, compared to inhibitor-free reference sample.”

- Please include a subsection in the introduction as a research significance to explain the novelty of your work.

-Thanks for the comments. Admixed organic corrosion inhibitor is one of the most efficient strategies to enhance corrosion resistance and durability of reinforced concrete. The inhibition mechanisms are closely related to chemical adsorption through the stable covalent bond by sharing lone pair electrons or π electrons between polar groups (e.g., N, O and P) or heterocyclic in organic corrosion inhibitors and vacant d orbitals of the reinforcement and physical adsorption through electrostatic attraction/Van der Waals force. Therefore, it can be found that the interaction between the traditional admixed organic corrosion inhibitor and reinforcement on its corrosion inhibition behaviors was strongly emphasized in previous studies. However, the optimization effect of the admixed organic corrosion inhibitor on concrete matrix, especially on pore structure and impermeability of concrete, was less reported. Therefore, a subsection as a research significance to explain the novelty of this present work was added into Introduction section in the revised manuscript as follows: “Based on the above literature review, it can be found that the interaction between the traditional admixed organic corrosion inhibitor and reinforcement on its corrosion inhibition behaviors was strongly emphasized in the previous studies. However, the optimization effect of the admixed organic corrosion inhibitor on concrete matrix, especially on pore structure and impermeability of concrete, was less reported. This study belongs to a research project about design and preparation of novel hydrophobic functional organic corrosion inhibitor for corrosion protection of reinforced concrete serving in extremely harsh marine environment. The proposed organic corrosion inhibitor on one hand can strongly adsorb on the reinforcement surface and subsequently improve corrosion resistance of the reinforcement; on the other hand, it can reduce the porosity and increase hydrophobicity of concrete, halting chloride transport in concrete matrix. The previous study [18] indicated that the prepared hydrophobic functional organic corrosion inhibitor effectively reduced corrosion rate of the reinforcement in SPS (containing 0.1 mol/l chlorides): after 168 h immersion, the polarization resistance (Rp) of the reference sample was about 50.3 kΩ·cm2; Rp increased to 77.1, 131.5, 192.4 and 213.6 kΩ·cm2 with 0.2, 0.4, 0.6 and 0.8 wt. % inhibitor in SPS, respectively. However, it is essential to clarify the influence of hydrophobic functional organic corrosion inhibitor on micro/macro properties of concrete matrix and corrosion performance of the reinforcement embedded in concrete. To this end, this present study aims at evaluating the inhibition effect of the novel hydrophobic functional organic corrosion inhibitor in reinforced concrete. On one hand, the influence of hydrophobic functional organic corrosion inhibitor on micro/macro properties of concrete was investigated, including contact angle, pore structure, water absorption behavior of concrete. On the other hand, the electrochemical behavior of the reinforcing steel embedded in concrete was characterized by OCP, electrochemical impedance spectroscopy (EIS) and potentio-dynamic polarization (PD).”

- Please include a brief but critical review regarding the conducted research studies in the introduction.

-Thanks for the comments. The inhibition mechanisms of the traditional organic corrosion inhibitor for the reinforcement can be described as the adsorption of organic molecules on the steel surface. Based on the above chemical/physical adsorption, the organic corrosion inhibitors form dense protection layer on the reinforcement surface, thus efficiently prevent the adsorption and contact of oxygen and chlorides, increasing the critical chloride concentration for corrosion initiation and reducing corrosion damage rate of the reinforcement. However, the influence of the traditional admixed organic corrosion inhibitor on concrete matrix was less reported, especially on pore structure and impermeability of concrete. According to the reviewer’s comment, a brief critical review regarding the conducted research studies was added into Introduction section in the revised manuscript as follows: “Based on the above literature review, it can be found that the interaction between the traditional admixed organic corrosion inhibitor and reinforcement on its corrosion inhibition behaviors was strongly emphasized in the previous studies. However, the optimization effect of the admixed organic corrosion inhibitor on concrete matrix, especially on pore structure and impermeability of concrete, was less reported.”

- Please add more explanation on the proposed methodology, verification process, and limitations.

-Thanks for the comments. According to the reviewer’s suggestion, more explanation on the proposed methodology, verification process, and limitations was supplemented and added into revised manuscript as follows:

“MIP is widely adopted for investigating the porosity and pore size distribution of cement-based materials, i.e. cement paste, mortar and concrete. Many factors affect the accuracy of MIP results, mainly including sampling, sample conditioning, pressure application rate, maximum applied intrusion pressure, used contact angle and surface tension of mercury. In addition, expansion of sample cell under pressure, differential mercury compression, sample compression and hydrostatic head of mercury also affect the accuracy of MIP results. The parameter used for MIP tests in this study is consistent with the reported studies to guarantee the accuracy of the derived results.”

“The contact angle is used to evaluate the affinity between the solid surface and liquid it is contacting: a smaller contact angle represents better surface wettability. The contact angle greater than 90° is corresponding to the solid surface which is not wettable by the liquid. Otherwise, the contact angle less than 90° is corresponding to the surface presenting good wettability with the formation of a larger solid–liquid interface.”

“The specimen weight was measured at an interval of 24 h until the fully dried state was achieved (about 7d to dry), i.e. the weight loss of the sample within 24 h was lower than 0.02 wt. %. Afterwards, the other 5 surfaces were isolated by epoxy and only one surface was exposed to water during capillary water absorption measurement to ensure one-way water transport [21]. The top surface was loosely covered with a piece of plastic film to avoid water evaporation. The exposed surface was immersed in water with about 5 mm depth and the weight change of the sample was recorded at different time intervals. The weight difference before and after water absorption is the cumulative water absorption quality of the sample at specific time interval. For each measurement, the specimen was quickly taken out, wiped with a dry towel to remove the free water on the surface, and then placed on the balance with the wet side up. The specimen was put back in water immediately to continue the water absorption test after weighing. The weighing process was completed within 15 s.”

- Please add more detail on the reported outcomes in Figure 3. Please add statistical characteristics of the graph accordingly.

-Thanks for the comments. Generally, the total porosity was proportional to the cumulative mercury intrusion. It was shown in Figure 3 (a) that no obvious peak was observed when pore diameter was larger than 150 nm, indicating that all pores in cement paste was less than 150 nm. In additional, a higher dosage of corrosion inhibitor was related to a lower cumulative mercury intrusion. As shown in Figure 3 (b), the critical pore diameter was 77.1 nm, 69.0 nm, 55.8 nm and 40.0 nm for the sample with 0, 1, 2 and 4 wt. % dosage of hydrophobic functional corrosion inhibitor, respectively. According to the reviewer’s comment, the above details about Figure 3 were added in the related paragraph and the relevant text was modified in the revised manuscript as follows: “Total porosity and pore size distribution of cement paste with different dosage of hydrophobic functional corrosion inhibitor at 28 days curing age are presented in Figure 3. It was shown in Figure 3 (a) that no obvious peak was observed when pore diameter was larger than 150 nm, indicating that all pores in cement paste was less than 150 nm. In additional, a higher dosage of corrosion inhibitor was related to a lower cumulative mercury intrusion. The total porosity of reference sample was about 0.161 ml/g and reduced to 0.153 ml/g with 1 wt. % dosage of hydrophobic functional corrosion inhibitor (Figure 3 (a)). When the dosage of corrosion inhibitor increased to 2 and 4 wt. %, the total porosity was further reduced to 0.142 and 0.139 ml/g, respectively. It indicated that the pore structure of cement paste was refined by hydrophobic functional corrosion inhibitor. As shown in Figure 3 (b), similar pore size distribution was relevant to both reference sample and inhibitor-containing sample. However, the peak corresponding to large pores (at about 100 nm) was shifted to smaller pore size and the corresponding peak intensity was also reduced by hydrophobic functional corrosion inhibitor. The critical pore diameter was 77.1 nm, 69.0 nm, 55.8 nm and 40.0 nm for the sample with 0, 1, 2 and 4 wt. % dosage of hydrophobic functional corrosion inhibitor, respectively.” The cumulative mercury intrusion amount for different samples were also marked in Fig.4 (a) in the revised manuscript.

- Please discuss the shortcomings of the conducted capillary water absorption test.

-Thanks for the comments. The capillary water absorption plays a dominant role in the ingress of corrosive medium in unsaturated porous cement-based materials and it is an important factor influencing the long-term durability of concrete structures. The advantage for capillary water absorption test exists in that this measurement is quite easy to be conducted in laboratory using the gravimetric technique. However, the shortcomings of the capillary water absorption test can be described as follows: first, the penetration depth and distribution of water molecules are unable to be obtained by this method. Further the influence of gravity on capillary absorption coefficient should also be considered in long-term experiments. According to the reviewer’s comment, the above advantage and shortcomings of capillary water absorption test were added in the revised manuscript as follows: “The advantage for capillary water absorption test exists in that this measurement is quite easy to be conducted in laboratory using the gravimetric technique. The shortcomings of the capillary water absorption test can be described as follows: first, the penetration depth and distribution of water molecules are unable to be obtained by this method. Further the influence of gravity on capillary absorption coefficient should also be considered in long-term experiments.”

- Please elaborate on the effect of adding more than 4% water content on the reported outcomes in Figure 8.

-Thanks for the comments. Figure 9 presents OCP of the reinforcement in concrete specimen immersed in 3.5 wt. % NaCl solution. It’s obvious that OCP of the reinforcement embedded in concrete admixed with hydrophobic functional organic corrosion inhibitor was significantly more positive, compared to the reference sample. OCP of the reinforcement with higher concentration inhibitor was slightly negative than the reinforcement with lower concentration inhibitor, but still more positive than the reference sample. Higher dosage of hydrophobic functional organic corrosion inhibitor was not relevant in this present study and the related reasons are as follows: first, the hydrophobic functional organic corrosion inhibitor with the dosage of 4 wt. % exhibited very high inhibition efficiency of 96 %, thus the authors believe that it is not necessary to further increase the dosage of hydrophobic functional organic corrosion inhibitor in this study. Further, as shown in the figure below, the admixed hydrophobic functional organic corrosion inhibitor retarded hydration rate of cement paste and the retardation effect was more pronounced with higher dosage. As a result, if the dosage of hydrophobic functional organic corrosion inhibitor is further increased, its retardation effect on hydration process of cement-based materials will be more severe, which is harmful for the mechanical property of concrete.

(a) Heat release rate               (b) Accumulative heat release amount

Fig.1 Heat release curve of cement paste with different dosage of hydrophobic functional organic corrosion inhibitors (w/c=0.5)

- Please add statistical analyses to further discuss the proposed method in the discussion. In addition, please add a comparative discussion on the most important parameters that can affect the presented reports in this study.

-Thanks for the comments. According to the reviewer’s comment, Discussions section was added in the revised manuscript as follows:

“3.5.4 Discussions

Based on the experimental results in this study, hydrophobic functional organic corrosion inhibitor significantly altered the performance of concrete and embedded reinforcement. For concrete, the pronounced pore refinement was relevant to the specimen in the presence of hydrophobic functional organic corrosion inhibitor, evidenced by reduced total porosity and content of large pores (Figure 4 and Table 8). Further, the average water contact angle was increased to 59.4°-80.5° for cement paste and 16.3°-38.9° for concrete in the presence of hydrophobic functional organic corrosion inhibitor (Figure 5), indicating that the hydrophobic property of concrete was increased by the inhibitor. The refined pore structure and increased hydrophobic property reduced the capillary water absorption amount of concrete (Figure 7), which was beneficial for halting chloride transport in concrete and corrosion initiation of the reinforcement. Due to lower internal humidity and chloride content caused by the halted water and chloride transport, the open circuit potential of the reinforcement was more positive when embedded in concrete in the presence of hydrophobic functional organic corrosion inhibitor (Figure 9). Further, effective adsorption on the reinforcement surface was relevant to hydrophobic functional organic corrosion inhibitor (Figure 8), mainly reducing the anodic corrosion reaction rate and significantly improving the corrosion resistance of the reinforced concrete, evidenced by one order of magnitude higher charge transfer resistance and one order of magnitude lower corrosion current density of the embedded reinforcement (Figure 11 and Figure 13). As a result, the novel hydrophobic functional organic corrosion inhibitor exhibited very high inhibition efficiency (in the range of 71 %-96 %) and obviously the better corrosion inhibition effect was relevant to higher dosage of corrosion inhibitor in concrete (96 % in the presence of 4 wt. % inhibitor).

Table 9 Comparison on the inhibition efficiency between hydrophobic functional organic corrosion inhibitor in this study and corrosion inhibitors in the reported studies [27-34].

Type of corrosion inhibitor

NaCl

Inhibitor

Time / d

Efficiency / %

Reed leaves extract

3.5 %

0.5 %

180

77.0

Ginger extract

8 %

4 %

60

40.9

Ca(NO2)2

8 %

4 %

60

25.7

Kelp extract

8 %

4 %

60

52.8

EG/AgNPs

natural seawater

5 %

90

82.6

Rice husk ash

5 %

20 %

40

85.0

Triethanolamine

3.5 %

1 %

28

64.2

Triethanolamine + Ca(NO2)2

3.5 %

1 %

28

90.1

Guanidine

1 M

0.5 M

30

77.5

1,6-Hexamethylenediamine

1 M

1 M

30

84.5

3-Aminopropyltriethoxysilane

1 M

1 M

30

62.6

Ethanolamine

3.5 %

5 L/m3

270

95.0

Monoflurophosphate

3.5 %

2 %

270

50.0

This study

3.5 %

4 %

72

96.0

Table 9 shows the comparison on the inhibition efficiency between hydrophobic functional organic corrosion inhibitor and conventional corrosion inhibitors in the reported studies in concrete [27-34]. It can be observed in Table 9 that due to different composition and concentration of corrosion inhibitors together with different concentrations of chlorides in concrete, the reported inhibition efficiency exhibited the value in a very wide range in the previous study. However, hydrophobic functional organic corrosion inhibitor proposed in this present study exhibited very high inhibition efficiency in concrete, compared to other corrosion inhibitors in the reported studies. Meanwhile the excellent inhibition effect maintained for a very long time.”

- Please revise the conclusion to present a condensed version highlighting the main contributions of your work.

-Thanks for the comments. According to reviewer’s comment, the conclusion was modified in the revised manuscript as follows: “In this study, the corrosion performance of reinforced concrete in the presence of novel hydrophobic functional organic corrosion inhibitor was evaluated. Hydrophobic functional organic corrosion inhibitor reduced the total porosity and amount of large pores, leading to refined pore structure of cement paste. Further, the capillary water absorption amount of cement-based materials was significantly reduced due to the reduced porosity and enhanced hydrophobic property of cement-based materials by hydrophobic functional organic corrosion inhibitor.

The embedded reinforcement exhibited more positive open circuit potential and higher corrosion resistance in concrete admixed with hydrophobic functional organic corrosion inhibitor due to the halted water and chloride transport process and subsequently lower internal humidity of concrete. Further, hydrophobic functional organic corrosion inhibitor was effectively adsorbed on the reinforcing steel surface, improving the corrosion resistance of the embedded reinforcement. The better corrosion inhibition effect was relevant to higher dosage of corrosion inhibitor in concrete. Therefore, the novel hydrophobic functional organic corrosion inhibitor exhibited very high inhibition efficiency and can be potentially used for efficient corrosion protection of reinforced concrete under severe marine environment.”

Reviewer 2 Report

The reviewed article is quite well written and contains interesting and convincing results. Nevertheless, I believe that some amendments and additions should be made to it before publication. To make it easier for the authors to see the extent of the proposed changes, I have listed them below in a numbered list in the order they appear in the text.

1. In Table 4, the authors give a fineness modulus value of 42.1 which is obviously wrong.

2. In Table 5, the last two columns are copies of column two and three (redundant copies).

3. In section 2.2, the authors write that steel bars were placed in the concrete after casting. It would be more accurate to write that in the concrete specimens, not in the concrete. In addition, some schematic drawing showing the position of the mentioned bars would be useful.

4. In section 2.3.3, the authors wrote that the specimens were dried in a vacuum chamber until a constant mass was obtained. In principle, such information is sufficient, but I would like the authors to write how long approximately the specimens were dried. This may not be essential information, but I think I am not the only one who would like to know.

5. If, in equations (2) and (3), the authors used the letter w to denote the same quantity, it is not appropriate to write it once as a capital letter and once as a lowercase letter, as this creates confusion. And if they are two different parameters, the latter should be defined.

6. The diagram of Randles circuit in Figure 9 is illegible. Please correct it. Furthermore, in the schematic and in the description in the text, the authors have given the symbol Qdl, which in figure 10 is probably erroneously replaced by Ydl. The unit was also incorrectly described in this figure. According to NIST, the form 'Mho' as a designation for the siemens (S) unit of the SI unit system is unaccepted and it should be strictly avoided.

7. Wherever this is missing, please provide in the figures, in addition to the symbols and units, a verbal designation of the parameters that are shown on the individual axes of the graphs.

I believe that once the above-mentioned changes have been made, the article can be submitted for publication.

Author Response

Reviewer 2

The reviewed article is quite well written and contains interesting and convincing results. Nevertheless, I believe that some amendments and additions should be made to it before publication. To make it easier for the authors to see the extent of the proposed changes, I have listed them below in a numbered list in the order they appear in the text.

- Thank you very much for the comments, all of which were taken into consideration and the text is revised accordingly. In what follows is our response on a point-by-point basis.

  1. In Table 4, the authors give a fineness modulus value of 42.1 which is obviously wrong.

-Thanks for the comments. We apologize for this mistake. The fineness modulus is an index to characterize the coarseness and type of fine aggregates but not used to characterize coarse aggregates. According to the reviewer’s suggestion, Table 4 was modified and column of fineness modulus wad deleted in the revised manuscript. Mud content, apparent density, loose bulk density, close packing density and air-void of coarse aggregates were 0.3 %, 2733 kg/m3, 1405 kg/m3, 1518 kg/m3 and 9.1 %, respectively. The relevant modified table is also listed further below.

Table 1. The physical properties of coarse aggregates used in this study.

Mud content (%)

Apparent

Density (kg/m3)

Loose bulk density (kg/m3)

Close packing

Density (kg/m3)

Air-void (%)

0.3

2733

1405

1518

9.1

  1. In Table 5, the last two columns are copies of column two and three (redundant copies).

-Thanks for the comments. According to the reviewer’s comment, Table 5 was modified and the redundant columns were deleted in the revised manuscript. The relevant modified table is also listed further below.

Table 2. The gradation of coarse aggregates used in this study.

Screen Size (mm)

16.000

10.000

5.000

2.5000

Total residue (%)

0

7.4

98.7

100

  1. In section 2.2, the authors write that steel bars were placed in the concrete after casting. It would be more accurate to write that in the concrete specimens, not in the concrete. In addition, some schematic drawing showing the position of the mentioned bars would be useful.

-Thanks for the comments. According to the reviewer’s comment, the preparation of reinforced concrete specimen in this present study was modified in the revised manuscript as follows: “The dimensions of cement paste and concrete was 4 cm ×4 cm ×16 cm and 10 cm ×10 cm ×10 cm, respectively. For reinforced concrete specimen, HPB235 carbon steel bar was centrally embedded in the specimen (the schematic diagram of reinforced concrete specimen in this study is shown in Figure 3). For concrete casting, cement and aggregates were first drying mixed by concrete mixer and the liquid hydrophobic functional organic corrosion inhibitor with different proportion was dispersed into tape water. The mixed aqueous solution was then poured into the above dry blend material and continuously stirred 3 minutes, followed by casting the fresh concrete into the mould. The top and bottom of the reinforcement were isolated with epoxy coating. The reinforcement was placed 2 cm from the edge of concrete specimens and the working length and surface area of the reinforcement electrode were 8 cm and 20.096 cm2.” The figure showing the schematic diagram of reinforced concrete specimens was also added in the revised manuscript and listed further below.

Figure 1. Schematic diagram of reinforced concrete specimens in this study.

  1. In section 2.3.3, the authors wrote that the specimens were dried in a vacuum chamber until a constant mass was obtained. In principle, such information is sufficient, but I would like the authors to write how long approximately the specimens were dried. This may not be essential information, but I think I am not the only one who would like to know.

-Thanks for the comments. According to the reviewer’s comment, the specimen drying process before water absorption tests was added into the revised manuscript as follows: “The specimen weight was measured at an interval of 24 h until the fully dried state was achieved (about 7 d), i.e. the weight loss of the sample within 24 h was lower than 0.02 %.”

  1. If, in equations (2) and (3), the authors used the letter w to denote the same quantity, it is not appropriate to write it once as a capital letter and once as a lowercase letter, as this creates confusion. And if they are two different parameters, the latter should be defined.

-Thanks for the comments. The authors apologize sorry for the mistake about letter w in equations (2) and (3). In this study, equations (2) was used to calculate the capillary water absorption amount per surface area of specimens at different time interval and equations (3) was used to calculate the capillary water absorption coefficient of specimens. Both in equations (2) and (3), the letter w denoted the specimen weight and symbol Δw(t) is the capillary water absorption amount per surface area at specific time interval (g/m2). According to the reviewer’s comment, the relevant text is the revised manuscript was modified as follows:

“The capillary water absorption amount per surface area at different time interval can be calculated by the following equation [21]:

Δw(t)=(wt-w0)/A                                      (2)

(2)

where Δw(t) is the capillary water absorption amount per surface area at specific time interval (g/m2); w0 is the initial sample weight before water absorption (g); wt is the sample weight after water absorption at specific time interval (g); A is the exposed surface area of cement paste or concrete sample (m2).

Subsequently, the capillary water absorption coefficient of cement or concrete sample can be calculated as follows:

k =Δw(t)·t-1/2                                               (3)

(3)

where k is the capillary water absorption coefficient (kg·m-2·h-1/2) and t is the absorption time (h).”

  1. The diagram of Randles circuit in Figure 9 is illegible. Please correct it. Furthermore, in the schematic and in the description in the text, the authors have given the symbol Qdl, which in figure 10 is probably erroneously replaced by Ydl. The unit was also incorrectly described in this figure. According to NIST, the form 'Mho' as a designation for the siemens (S) unit of the SI unit system is unaccepted and it should be strictly avoided.

-Thanks for the comments. According to the reviewer’s comment, Randles circuit used in this study as shown in Fig. 10 was modified in the revised manuscript as follows:

Figure 2. EIS of the reinforcement embedded in concrete immersed in 3.5 wt. % NaCl solution at different ages.

In addition, Ydl was incorrectly used in the schematic and should be replaced by Qdl. As a result, Ydl in Figure 11 was corrected to Qdl and its unit was also corrected to mΩ*s^/cm2 in the revised manuscript. The relevant figure is also listed further below.

Figure 3. The fitted Rct and Qdl of the reinforcement embedded in concrete immersed in 3.5 wt. % NaCl solution at different ages.

  1. Wherever this is missing, please provide in the figures, in addition to the symbols and units, a verbal designation of the parameters that are shown on the individual axes of the graphs.

-Thanks for the comments. All provided figures are checked according to the Reviewer's suggestion. When the verbal designation of the parameters is necessary, it is marked in the graphs.

Reviewer 3 Report

The authors proposed the utilization of a novel hydrophobic functional organic corrosion inhibitor in reinforced concrete. The results shows that the hydrophobic functional organic corrosion inhibitor decreased both the porosity and the capillarity of cement pastes and reinforced concrete. Furthermore, the inhibitor exhibited good inhibition effect on the reinforcement embedded in concrete. According to study, the dosage of 4 wt.% presented the best results. The manuscript is very well structured, well organized, and easy to follow.

Specific comments:

 (1)   Particle size distribution: Please, use the term “Sieve size” instead of “screen size”. Is the total residue percent retained or passing through each sieve? Please, check both tables 3 and 5.

(2)   Section 2.2. Please, clarify why cement pastes and concrete were used in the study. Moreover, were cement pastes, concrete, and reinforcement concrete used? Explain the methodology used to prepare the samples, mixing procedure, form of inhibitor application, etc.

(3)   Figure 3: It is difficult to distinguish curves 2 wt.% and 4 wt.%, because both lines look similar on the graphic. A suggestion, use one dotted line and another dashed line.

(4)   Page 7: Lines 235 – 245. Can you explain why did the contact angle presented highly dispersed results in each sample? In concrete it is easy to understand this behavior due to heterogeneity.

(5)   If total porosity is close in the samples with 2 wt. % and 4 wt.% (0.142 and 0.139 ml/g, respectively) and there is a pore refinement in the sample with 4 wt.%, why was the capillarity lower in the samples with 4 wt.%? In this case, can the hydrophobic inhibitor have any other physical effect besides the pore refinement?

(  (6) It is suggested that the authors link the results observed for different properties considered in this study with more bibliographic references. The discussion section must be further improved.

Author Response

Reviewer 3

The authors proposed the utilization of a novel hydrophobic functional organic corrosion inhibitor in reinforced concrete. The results shows that the hydrophobic functional organic corrosion inhibitor decreased both the porosity and the capillarity of cement pastes and reinforced concrete. Furthermore, the inhibitor exhibited good inhibition effect on the reinforcement embedded in concrete. According to study, the dosage of 4 wt.% presented the best results. The manuscript is very well structured, well organized, and easy to follow.

- Thank you very much for the comments, all of which were taken into consideration and the text is revised accordingly. In what follows is our response on a point-by-point basis.

Specific comments:

 (1)   Particle size distribution: Please, use the term “Sieve size” instead of “screen size”. Is the total residue percent retained or passing through each sieve? Please, check both tables 3 and 5.

-Thanks for the comments. According to the reviewer’s comment, “screen size” was replaced by “Sieve size” in the revised manuscript. In Tables 3 and Table 5, the total residue percent was referred to the fine aggregates / coarse aggregates retained on each sieve.

(2)   Section 2.2. Please, clarify why cement pastes and concrete were used in the study. Moreover, were cement pastes, concrete, and reinforcement concrete used? Explain the methodology used to prepare the samples, mixing procedure, form of inhibitor application, etc.

-Thanks for the comments. In fact, the pore structure was investigated with cement pastes to avoid the heterogenous of concrete caused by aggregates. The contact angle and capillary water absorption coefficient were investigated with both cement paste and concrete and similar trend was observed for the above two specimens. Reinforced concrete was only used to evaluate the effect of hydrophobic functional corrosion inhibitor on electrochemical behavior of reinforced concrete. For concrete casting, cement and aggregates were first drying mixed by concrete mixer and the liquid hydrophobic functional organic corrosion inhibitor with different proportion was dispersed into tape water. The mixed aqueous solution was then poured into the above dry blend material and continuously stirring 3 minutes, followed by casting the fresh concrete into the mould. According to the reviewer’s comment, the relevant text was added in Section 2.2 in the revised manuscript as follows: “The dimensions of cement paste and concrete was 4 cm ×4 cm ×16 cm and 10 cm ×10 cm ×10 cm, respectively. For reinforced concrete specimen, HPB235 carbon steel bar was centrally embedded in the specimen (the reinforced concrete specimen in this study is shown in Figure 3). For concrete casting, cement and aggregates were first drying mixed by concrete mixer and the liquid hydrophobic functional organic corrosion inhibitor with different proportion was dispersed into tape water. The mixed aqueous solution was then poured into the above dry blend material and continuously stirred 3 minutes, followed by casting the fresh concrete into the mould. For reinforced concrete, the top and bottom of the reinforcement were isolated with epoxy coating. The reinforcement was placed 2 cm from the edge of concrete specimens and the working length and surface area of the reinforcement electrode were 8 cm and 20.096 cm2.”

(3)   Figure 3: It is difficult to distinguish curves 2 wt.% and 4 wt.%, because both lines look similar on the graphic. A suggestion, use one dotted line and another dashed line.

-Thanks for the comments. According to the reviewer’s suggestion, the relevant figure was modified for better expression in the revised manuscript and also listed further below.

     (a) Total porosity

(b) Pore size distribution

Figure 1. Total porosity (a) and pore size distribution (b) of cement paste with different dosage of hydrophobic functional corrosion inhibitor at 28 days curing age.

(4)   Page 7: Lines 235 – 245. Can you explain why did the contact angle presented highly dispersed results in each sample? In concrete it is easy to understand this behavior due to heterogeneity.

-Thanks for the comments. It was indeed that the contact angle presented highly dispersed results in each cement paste sample. The related reasons can be explained as follows: the cement paste is a heterogeneous system, which is consisted of un-hydrated cement particles, different hydration products and pores with different size. As a result microstructural defects on the surface of cement paste sample, e.g. capillary pores and air voids, are inevitable, leading to inhomogeneity of the sample surface. During the contact angle tests, when water droplet spread on the surface of cement paste sample, it encountered the pining of rough microstructure. As a result, the apparent contact angle between water droplet and cement paste surface was no longer constant, but fluctuated in a certain range.

(5)   If total porosity is close in the samples with 2 wt. % and 4 wt.% (0.142 and 0.139 ml/g, respectively) and there is a pore refinement in the sample with 4 wt.%, why was the capillarity lower in the samples with 4 wt.%? In this case, can the hydrophobic inhibitor have any other physical effect besides the pore refinement?

-Thanks for the comments. The capillary water absorption amount of cement sample was significantly related to its porosity and hydrophobic property. In this study, the hydrophobic functional corrosion inhibitor could reduce the total porosity and content of large pores, leading to refined pore structure of cement paste. In addition, the reported inhibitor enhanced hydrophobic property of cement sample according to contact angle results and this positive effect was more pronounced with higher inhibitor concentration. Therefore, the samples with 4 wt.% inhibitor presented lower capillarity than samples with 2 wt.% inhibitor even both samples presented similar total porosity. In the revised manuscript, the above discussions were added as follows: “Based on contact angle results (Figure 5), the hydrophobic functional corrosion inhibitor enhanced hydrophobic property of cement/concrete sample and this positive effect was more pronounced with higher inhibitor concentration. Therefore, the cement paste and concrete samples with 4 wt.% inhibitor presented lower capillarity than samples with 2 wt.% inhibitor even both samples presented similar total porosity (Figure 4).”

(6) It is suggested that the authors link the results observed for different properties considered in this study with more bibliographic references. The discussion section must be further improved.

-Thanks for the comments. According to the reviewer’s suggestion, Discussion section was added in the revised manuscript as follows:

“3.5.4 Discussions

Based on the experimental results in this study, hydrophobic functional organic corrosion inhibitor significantly altered the performance of concrete and embedded reinforcement. For concrete, the pronounced pore refinement was relevant to the specimen in the presence of hydrophobic functional organic corrosion inhibitor, evidenced by reduced total porosity and content of large pores (Figure 4 and Table 8). Further, the average water contact angle was increased to 59.4°-80.5° for cement paste and 16.3°-38.9° for concrete in the presence of hydrophobic functional organic corrosion inhibitor (Figure 5), indicating that the hydrophobic property of concrete was increased by the inhibitor. The refined pore structure and increased hydrophobic property reduced the capillary water absorption amount of concrete (Figure 7), which was beneficial for halting chloride transport in concrete and corrosion initiation of the reinforcement. Due to lower internal humidity and chloride content caused by the halted water and chloride transport, the open circuit potential of the reinforcement was more positive when embedded in concrete in the presence of hydrophobic functional organic corrosion inhibitor (Figure 9). Further, effective adsorption on the reinforcement surface was relevant to hydrophobic functional organic corrosion inhibitor (Figure 8), mainly reducing the anodic corrosion reaction rate and significantly improving the corrosion resistance of the reinforced concrete, evidenced by one order of magnitude higher charge transfer resistance and one order of magnitude lower corrosion current density of the embedded reinforcement (Figure 11 and Figure 13). As a result, the novel hydrophobic functional organic corrosion inhibitor exhibited very high inhibition efficiency (in the range of 71 %-96 %) and obviously the better corrosion inhibition effect was relevant to higher dosage of corrosion inhibitor in concrete (96 % in the presence of 4 wt. % inhibitor).

Table 9 shows the comparison on the inhibition efficiency between hydrophobic functional organic corrosion inhibitor and conventional corrosion inhibitors in the reported studies in concrete [27-34]. It can be observed in Table 9 that due to different com-position and concentration of corrosion inhibitors together with different concentrations of chlorides in concrete, the reported inhibition efficiency exhibited the value in a very wide range in the previous study. However, hydrophobic functional organic corrosion inhibitor proposed in this present study exhibited very high inhibition efficiency in concrete, compared to other corrosion inhibitors in the re-ported studies. Meanwhile the excellent inhibition effect maintained for a very long time.”

The table showing the comparison on the inhibition efficiency between hydrophobic functional organic corrosion inhibitor in this present study and conventional corrosion inhibitors in the reported studies in concrete was also added in the revised manuscript and listed further below.

Table 1 Comparison on the inhibition efficiency between hydrophobic functional organic corrosion inhibitor in this study and corrosion inhibitors in the reported studies [27-34].

Type of corrosion inhibitor

NaCl

Inhibitor

Time / d

Efficiency / %

Reed leaves extract

3.5 %

0.5 %

180

77.0

Ginger extract

8 %

4 %

60

40.9

Ca(NO2)2

8 %

4 %

60

25.7

Kelp extract

8 %

4 %

60

52.8

EG/AgNPs

natural seawater

5 %

90

82.6

Rice husk ash

5 %

20 %

40

85.0

Triethanolamine

3.5 %

1 %

28

64.2

Triethanolamine + Ca(NO2)2

3.5 %

1 %

28

90.1

Guanidine

1 M

0.5 M

30

77.5

1,6-Hexamethylenediamine

1 M

1 M

30

84.5

3-Aminopropyltriethoxysilane

1 M

1 M

30

62.6

Ethanolamine

3.5 %

5 L/m3

270

95.0

Monoflurophosphate

3.5 %

2 %

270

50.0

This study

3.5 %

4 %

72

96.0

Round 2

Reviewer 1 Report

N/A